# Beach Leveling Using a Remotely Piloted Aircraft System (RPAS): Problems and Solutions



**Francisco Contreras-de-Villar [1],\*, Francisco J. García [2], Juan J. Muñoz-Perez [1], Antonio Contreras-de-Villar [1], Veronica Ruiz-Ortiz [1], Patricia Lopez [1], Santiago Garcia-López [3] and Bismarck Jigena [4]**

1 Department of Industrial Engineering and Civil Engineering, Campus Bay of Algeciras, University of Cadiz, Avda. Ramón Puyol s/n, 11202 Algeciras, Spain; juanjose.munoz@uca.es (J.J.M.-P.); antonio.contreras@uca.es (A.C.-d.-V.); veronica.ruiz@uca.es (V.R.-O.); patricia.lopezgarcia@uca.es (P.L.)
2 Urbing-Lab Diseño y Gestión, C/Cabo Noval 6, 52005 Melilla, Spain; urbing.lab@gmail.com
3 Department of Earth Sciences, Campus Río San Pedro s/n, University of Cadiz, 11510 Puerto Real, Spain; santiago.garcia@uca.es
4 Department of Nautical Sciences and Maritime Studies, Campus Río San Pedro s/n, University of Cadiz, 11510 Puerto Real, Spain; bismarck.jigena@uca.es
\* Correspondence: francisco.contreras@uca.es

**Abstract:** The size and great dynamism of coastal systems require faster and more automated mapping methods like the use of a remotely piloted aircraft system (RPAS) or unmanned aerial vehicle (UAV). This method allows for shorter intervals between surveys. The main problem for surveying using low-altitude digital photogrammetry in beach areas is their visual homogeneity. Obviously, the fewer the homologous points defined by the program, the lower the accuracy. Moreover, some factors influence the error performed in photogrammetric techniques, such as flight height, flight time, percentage of frame overlap (side and forward), and the number of ground control points (GCPs). A total of 72 different cases were conducted varying these factors, and the results were analyzed. Among the conclusions, it should be highlighted that the error for noon flights is almost double that for the early morning flights. Secondly, there is no appreciable difference regarding the side overlap. But, on the other side, RMSE increased to three times (from 0.05 to 0.15 m) when forward overlap decreased from 85% to 70%. Moreover, relative accuracy is 0.05% of the flying height which means a significant increase in error (66%) between flights performed at 60 and 100 m height. Furthermore, the median of the error for noon flights (0.12 m) is almost double that for the early morning flights (0.07 m) because of the higher percentage of grids with data for early flights. Therefore, beach levelings must never be performed at noon when carried out by RPAS. Eventually, a new parameter has been considered: the relationship between the number of GCPs and the surface to be monitored. A minimum value of 7 GCP/Ha should be taken into account when designing a beach leveling campaign using RPAS.

**Keywords:** UAV; RPAS; littoral systems; aerial photogrammetry; DTM; monitoring; SfM; GCPs

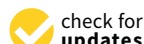



## 1. Introduction

Coastal erosion has become one of the most important concerns of different countries [1]. Coastal areas are the focal points of tourist attractions, which translates into an important source of economic income [2–4]. Moreover, the study of coastal behavior helps us understand the complex processes that occur in these areas [5,6]. Their understanding leads us to the prevention of coastal erosion, and thus monitoring the evolution of our beaches is essential [6]. Thus, a methodology for carrying out measurements of some oceanographic phenomena using Unmanned Aerial Vehicles (UAV also known as remotely piloted aircraft system or RPAS) have already been presented by other researchers (e.g., [7]). Nevertheless, some aspects can still be taken into account as we will show later.

Correct coastal modeling needs a three-dimensional reconstruction of the study area [8]. Coastal modeling is represented by digital terrain models (DTMs) of high spatial

resolution. Geomorphological state can be defined as a multitemporal surface [9,10]. Depending on the beach area to be mapped (dry zone, intertidal zone, or submerged zone), various techniques and methodologies can be used [11,12]. The dry beach and the intertidal zone have been mapped using direct topography techniques [12]. Initially, a tachymeter was used, being later replaced by the total station, an electronic transit theodolite integrated with electronic distance measurement (EDM), and an on-board computer to collect data and perform triangulation calculations. This task is currently done with GPS techniques. This type of point-to-point data collection is cheaper than the previous ones because it only requires one technician. However, GPS surveying is limited to small and easily accessible areas.

The size and great dynamism of these coastal systems require faster and more automated mapping methods [13]. Thus, the synchronous nature of the data is not lost [14]. Photogrammetry has evolved to the technique called structure from motion (SfM) [13–17] based on algorithms that allows one to obtain excellent cartographic results from a set of frames that cover an area. The emergence of RPAS systems as well as the high definition of today's digital cameras have induced new cartographic systems. The use of these systems significantly reduces costs and execution times, providing excellent accuracy [18]. This technology, tested in multiple applications, appears as a serious competitor against other cartographic techniques (e.g., LIght Detection and Ranging or LIDAR) [19]. Hugengoltz et al. [20], for instance, stated that the vertical RMSE of an RPAS data set was equivalent to the RMSE of a bare earth LiDAR DTM for the same site.

The work procedure involves the definition of a series of parameters such as flight height (Figure 1a), covering area, percentage of overlap between adjacent frames (Figure 1b,c), and a different number of ground control points (GCPs) [21] (Figure 1d).

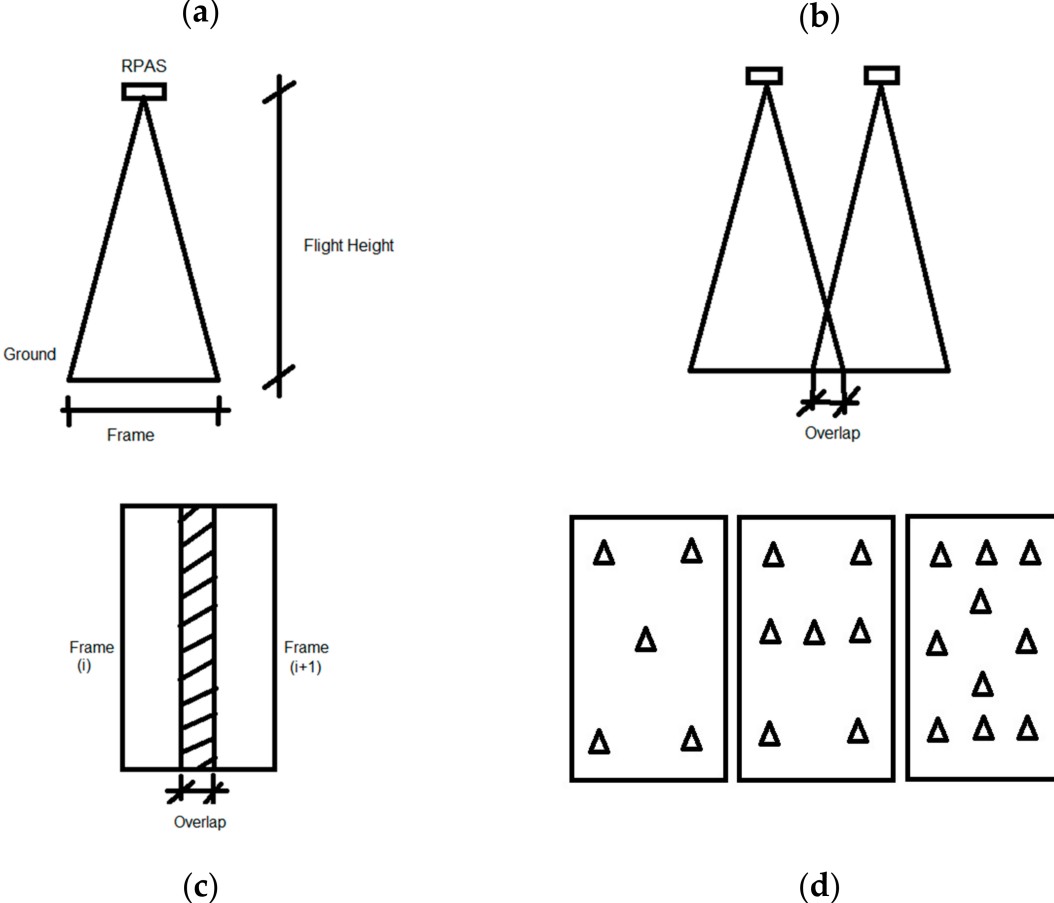

**Figure 1.** Sketch showing basic concepts of the remotely piloted aircraft system (RPAS) system: frame and flight height (**a**), overlap between two adjacent frames (**b**,**c**) and distribution of the different number of ground control points (GCPs) (**d**).

A particular case of the problems of low-altitude digital photogrammetry is the identification of common points in contiguous frames over poorly differentiated visual areas. When performing a low-altitude flight over highly homogeneous surfaces (beach sand, snow, agricultural areas of the same crop) is difficult to find common points [22]. The fewer the homologous points defined by the program, the lower the accuracy. This fact is common in the photogrammetry of beach areas and the accuracy of the DTM will be the result of the concatenation of the errors in different stages [23].

Thus, this paper aims to compare the vertical accuracy of a beach leveling, by using an RPAS, performed with different parameters of flight (height, time, side and forward overlap) and number of GCPs. Eventually, some guidelines will be presented to minimize the error of a photogrammetric survey.

## 2. Study Area

The chosen beach is Los Lances beach, in Tarifa (SW Spain). This area is considered as a bird special protection area due to its privileged situation that gives it a relevant role in air and marine migration processes (Figure 2).

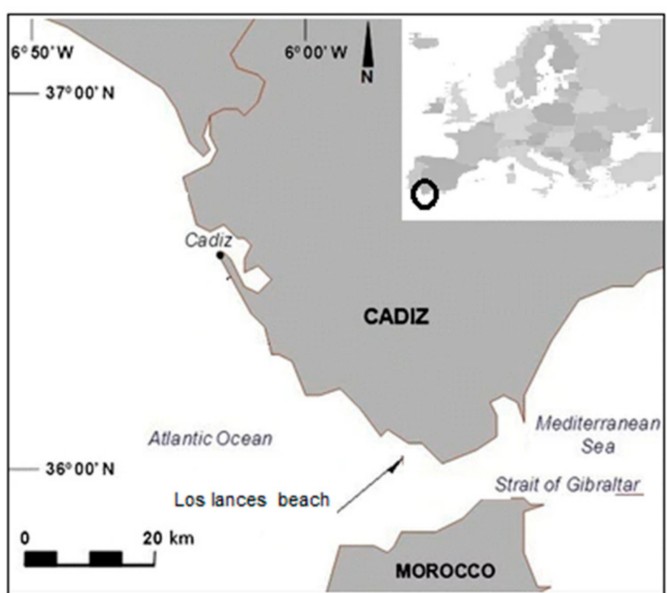

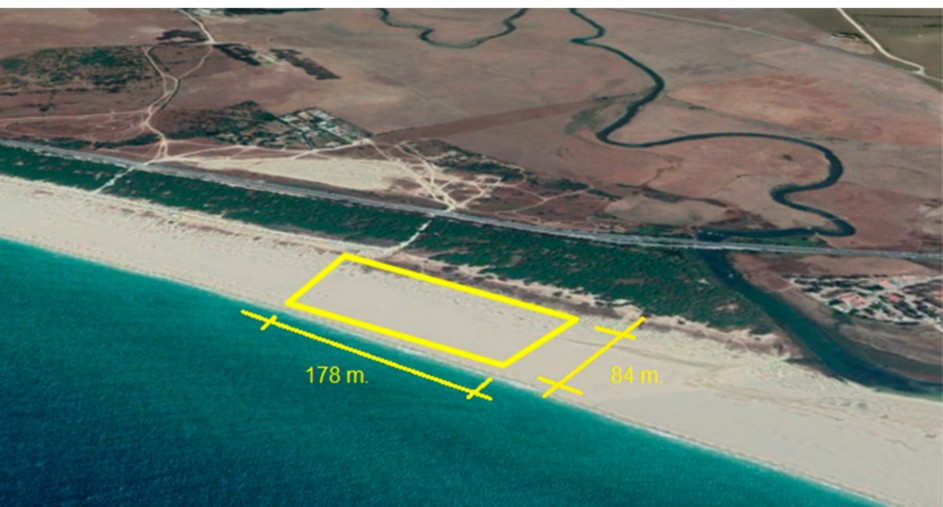

**Figure 2.** Location of the study area.

This space has a good state of conservation of ecosystems and a high-quality landscape. The beach is 3854 m long and covers an area of 280,000 m$^2$. The study area, a 178 m by 84 m rectangle, is also shown in Figure 2.

Its degree of urbanization is low. It has fine golden-colored sand, composed of medium-coarse unconsolidated sediments. The D$_{50}$ of the emerged sand is 0.34 mm. It is a dissipative beach and has waves of medium-moderate degree [24]. The maximum tidal range is about 1.40 m, and the significant wave height Hs is about 3.7 m [21]. It is a type of semi-urban beach widely used by windsurfers and kitesurfers due to the abundant windy days of the year that occur in this area. The prevailing winds in the area are eastwards and westwards [25]. The dry beach area before the dune area is over 100 m wide, which is an optimal area for conducting the study.

## 3. Methods

The factors that influence the error performed in digital photogrammetric techniques are flight height, overlap (side and forward), and GCP number. Different tests were conducted varying these factors, and the results were analyzed. The main problem for surveying using photogrammetric methods in beach areas is their visual homogeneity. This effect reduces the number of homologous points among neighboring frames and thus prevents optimal correlation.

In normal conditions, it is recommended to make the flights in hours close to noon since the sun is in the highest position, generating few shadows [26]. However, in addition to the noon flights, other flights were carried out early in the morning when the sun was at a low altitude. These supplementary flights were performed trying to find out if the shadows produce enough differentiation in the terrain as to decrease the margin of error (Figure 3).

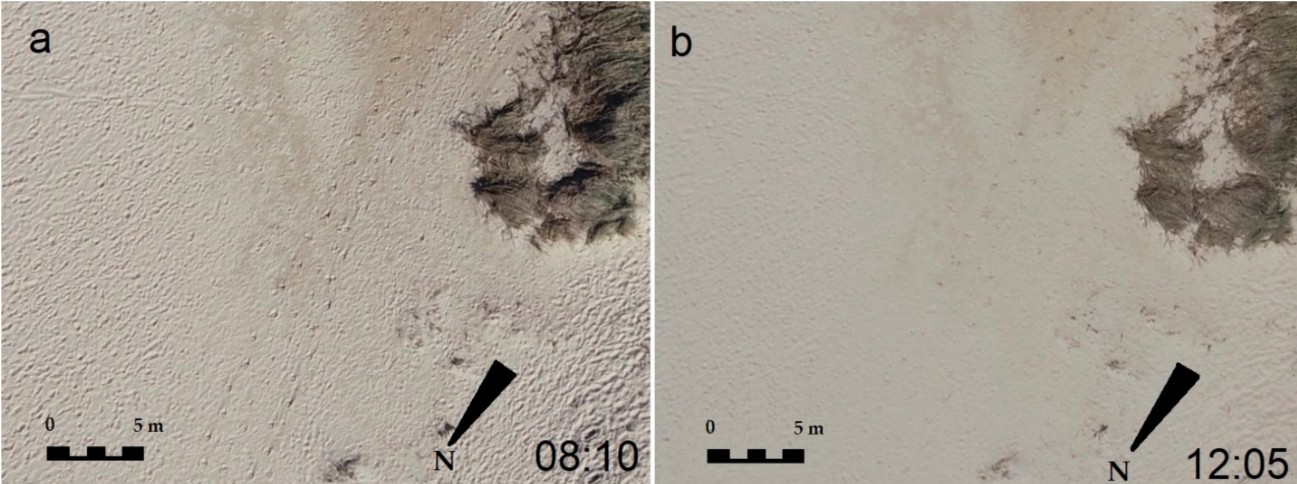

**Figure 3.** Frame detail: (**a**) early morning; (**b**) at noon. Frame details showing how shadows produce more homologous points in the early morning than at noon.

One difficulty is the short flight time of RPAS. Thus, the usual flight time (20–25 min) must be balanced against each other parameter: the surface to be flown, the flight height, and the overlap in the images we want to obtain.

The parameters that varied on each flight were the following:

- Data collection at 8 a.m. and 12 p.m.,
- Flight height at 60, 80, and 100 m,
- Side overlap at 85% and 70%,
- Forward overlap at 85% and 70%,
- Number of GCPs on each flight: 10, 7, and 5.

### 3.1. Data Collection

As we previously specified, the area chosen for the study is 178 m alongshore and 84 m cross-shore, and therefore its surface area is about 15,000 m², although the overflight area was obviously taken of a larger surface.

Data collection was planned with Phantom 4 Pro, based on the following three stages:

(a) Distribution on the beach of prefabricated landmarks to improve precision and calibration of the camera. Though some authors [27] state that direct georeferencing with high camera location accuracy and GNSS receivers can limit the need for GCPs, these landmarks were used as GCPs and georeferenced. The GCPs were plastic, measuring 24 × 30 cm and about 5 mm high (Figure 4), and had a hollow that helped fix their position in the sand. The reverse was painted, creating an alternating white-and-black grid. The positions were chosen to try to get an optimal placement according to the literature (covering all four corners of the site, the highest and lowest elevations, and with sufficient cross-shore and alongshore coverage). Moreover, their location was maintained during the two flight campaigns so that the results were not affected by any movement.

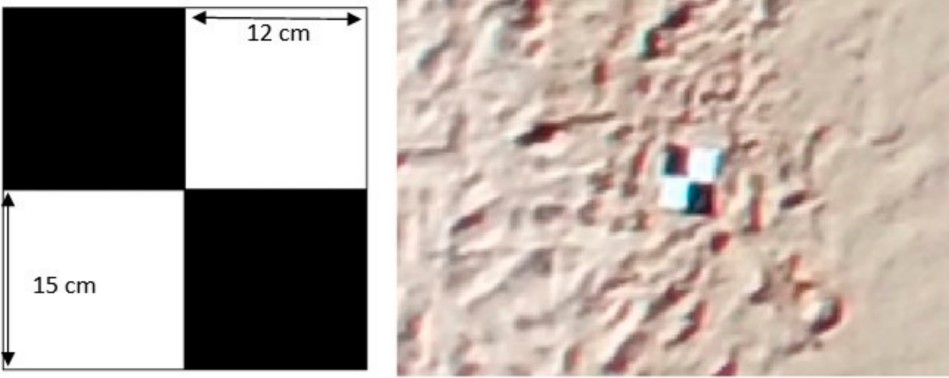

**Figure 4.** Dimensions of the GCP used and an aerial view of one of them already placed on the beach.

(b) Performing a topographic survey of the area using direct topography with GPS in RTK (Real-Time Kinematic) mode that provides precision around 3 cm. To ensure this accuracy, each topographic reading was repeated by taking three consecutive shots, which were checked and validated only if their difference was less than 1 cm in planimetry and 2 cm in altimetry. Moreover, the GPS rod man distinguished all the locations where pronounced changes in the beach topography appeared, due to his/her training and experience. Therefore, the density of GPS points was increased in these areas. A total of 657 survey points were taken, with an average distance among points of five meters approximately. This density of points is very high for the characteristics of the terrain.

The aims were twofold: first, calculating a topographic surface to compare with the photogrammetric data and, second, determining the coordinates of the GCPs for the RPAS postprocessing. The points were defined in European coordinates UTM ETRS89, and the levelings referred to the Spanish Datum (mean sea level in Alicante) by using the EGM2008 geoid provided by the National Geographic Institute [28].

(c) Introduction of the flight parameters into the RPAS software and realization of the photogrammetric flights. Six flights were made on the same day. In this way, the weather conditions and the situation of the terrain would be the same and therefore would not influence the results of the study. Flight planning requires the establishment of the limits of the area to be photographed, camera characteristics, flight height, flight direction, and frame overlap in side and forward directions. Thus, three flights were made at 8 am (when the sun angle is still low), with flight heights of 60, 80, and 100 m. The flights were taken with a side and forward overlap of 85%. However, also 75% of overlap in both directions was considered afterward by using the software. There were three more flights at noon (when the sun is at its highest position), repeating the same operation.

Therefore, six flights were made, but a total of 72 cases were studied (Table 1) by combining the parameters of two times of the day with different heights of flight (3), different side and forward overlaps and different number of GCPs (5, 7, and 10).

**Table 1.** Values of the different flight parameters and number of studied cases.

| Parameter | Values | Number of Cases |
|---|---|---|
| Flight time | 8 a.m. and 12 p.m. | 2 |
| Flight height | 60, 80, and 100 m | 3 |
| Longitudinal overlap | 70% and 85% | 2 |
| Transverse overlap | 70% and 85% | 2 |
| Number of GCPs | 5, 7, and 10 | 3 |
| Total number of cases | - | 72 |

Flight mission planning was previously done. For this, the Pix4D Capture program was used. This program uses the aerial images of Google Earth® as a base on which the area to be flown is defined, the flight height and the side and forward coverings were described, and the flight course was marked to optimize the times. The program calculated the flight speed and shooting interval among photographs. Figure 5 shows the flight plan scheme.

The camera technical data are presented in Table 2.

**Table 2.** Data of camera.

| Sensor | 1″ CMOS |
|---|---|
| | **Effective Pixels: 20M** |
| Lens | FOV 84° 8.8 mm/24 mm (35 mm format equivalent) f/2.8−f/11 autofocus at 1 m-∞ |
| Iso Range | Photo: |
| | 100–3200 (Auto) |
| | 8–1/2000 s |
| | 8–1/8000 s |
| Mechanical Shutter Speed | 3:2 Aspect Ratio 5472 × 3648 |
| Electronic Shutter Speed | 4:3 Aspect Ratio 4864 × 3648 |
| Image Size | 16:9 Aspect Ratio 5472 × 3078 |
| | 4096 × 2160 (4096 × 2160 24/25/30/48/50p) |
| | 3840 × 2160 (3840 × 2160 24/25/30/48/50/60p) |
| PIV Image size | 2720 × 1530 (2720 × 1530 24/25/30/48/50/60p) |
| | Single Shot |
| | Burst Shooting: 3/5/7/10/14 frames |
| Still Photography modes | Auto Exposure Bracketing (AEB): dL/5 at 0.7 |
| | EV Bias Internal 2/3/5/7/10/15/20/30/60 s |

Number of frames and duration of flight are shown in Table 3.

**Table 3.** Flight data, frame characteristics, frame number, and duration of flight.

| Flight Height (m) | GSD (cm/pixel) | Frame Size (m·m) | Number of Frames | Duration of Flight (min) |
|---|---|---|---|---|
| 60 | 1.64 | 90 × 60 | 185 | 16 |
| 80 | 2.18 | 119 × 80 | 123 | 11 |
| 100 | 2.73 | 148 × 98 | 81 | 9 |

The second column displays the Ground Sampling Distance (GSD), which is directly related to the flight altitude and camera parameters. The GSD is defined as the distance between two consecutive pixel centers measured on the ground. The greater the GSD value, the lower the spatial resolution of the image, and the less visible the details [29].

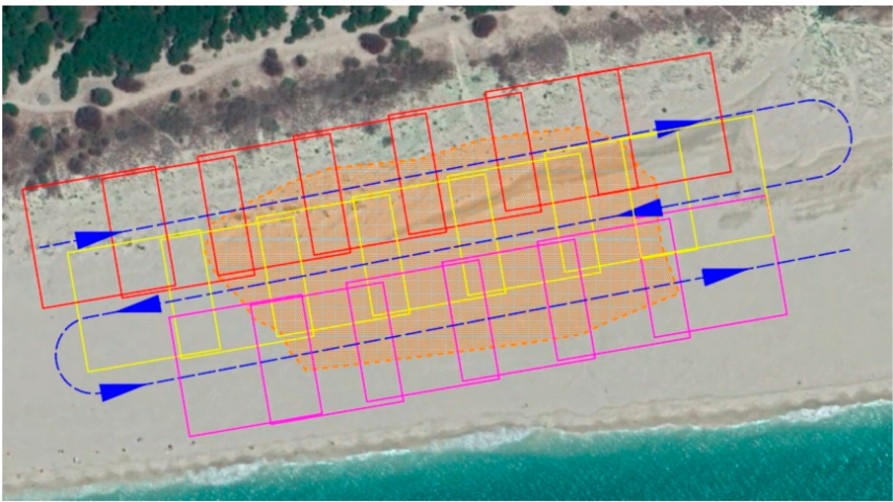

**Figure 5.** Image of the flight plan indicating the area of interest and frame overlap. The overflight area is bigger than the study area.

### 3.2. Method of Obtaining DTM by Photogrammetry and DTM Checking

The methodology for obtaining the DTM is based on the structure from motion (SfM) algorithm. The software used is Agisoft-Metashape Professional Educational®. Unstructured aerial images using fast, inexpensive, and highly automated image processing produces three-dimensional information. This RPAS-SfM pairing gives good results in cartographic production [27,30,31]

Firstly, once a set of frames was loaded into the software, an approximate orientation of the frames, based on the EXIF data of each photograph, was performed. EXIF is short for exchangeable image file, a format that is a standard for storing interchange information in digital photography image files using JPEG compression. It relied mainly on the focal length of the camera used, the time of taking the picture, and GPS coordinates.

Once the complete block was ordered, the program searched for tie points among adjacent frames. At this point, we could define the degree of precision that we require, as well as the key points and maximum tie points to be used in each frame to perform the operation.

The result of this process was a global point cloud that collected all the tie points of the flight frameset. At this time, the program had already created a three-dimensional point cloud. These point clouds were adjusted, georeferenced, and corrected for the lens distortion by using the GCPs. This procedure required entering the coordinates (X, Y, Z) of the GCPs and identifying them graphically in each of the frames in which they appeared. Since the GCPs points were defined in coordinates in the UTM-ETRS89 system, the adjusted point cloud would be in that same system.

At this point, we had the points of the topographic survey performed by GPS on the ground and the 72 DTMs obtained by processing the former set of point clouds. To facilitate and simplify the statistical reading of the DTMs, the size of each basic element of the DTM (tile size) was defined as a square of 1 m on the side. The Z value of each tile was defined as the average of the specific values it contained. Once the DTM was obtained, we cut it to the area of interest. By forming the DTM with all the points and cutting it later, we avoided the loss of data and extrapolation in the boundary areas.

These DTMs have been widely used, and much research on their error and uncertainty has already been investigated [14]. The quality of these models depends on several factors, such as the method used to attain the altimetric data, the density of the starting data, the resolution of the mesh, or the interpolation algorithm used, among others.

### 3.3. Calculation of the Error

To check the final quality of each flight, the error of each of the 72 DTMs (generated from the cloud of points obtained with the RPAS) was calculated by comparing to the DTM defined from the topographic data taken with GPS in RTK as the reference (Equation (1)). All the DTMs had the same dimensions, and a grid size of 1 × 1 m was chosen to facilitate comparison. Moreover, the percentage of grids that contained at least one datum was calculated, and its value was used as another reliability parameter.

The result of the comparison is another DTM whose characteristic is the difference between the altitudes of the flight DTM and the topographic DTM (GPS on the ground), that is, the vertical error ($\varepsilon$) in every grid.

$$\varepsilon = Z_{flight} - Z_{ground} \tag{1}$$

Figure 6 shows an example of this error ($\varepsilon$) calculated as the difference between the altitudes of the flight DTM and the topographic DTM (GPS on the ground)

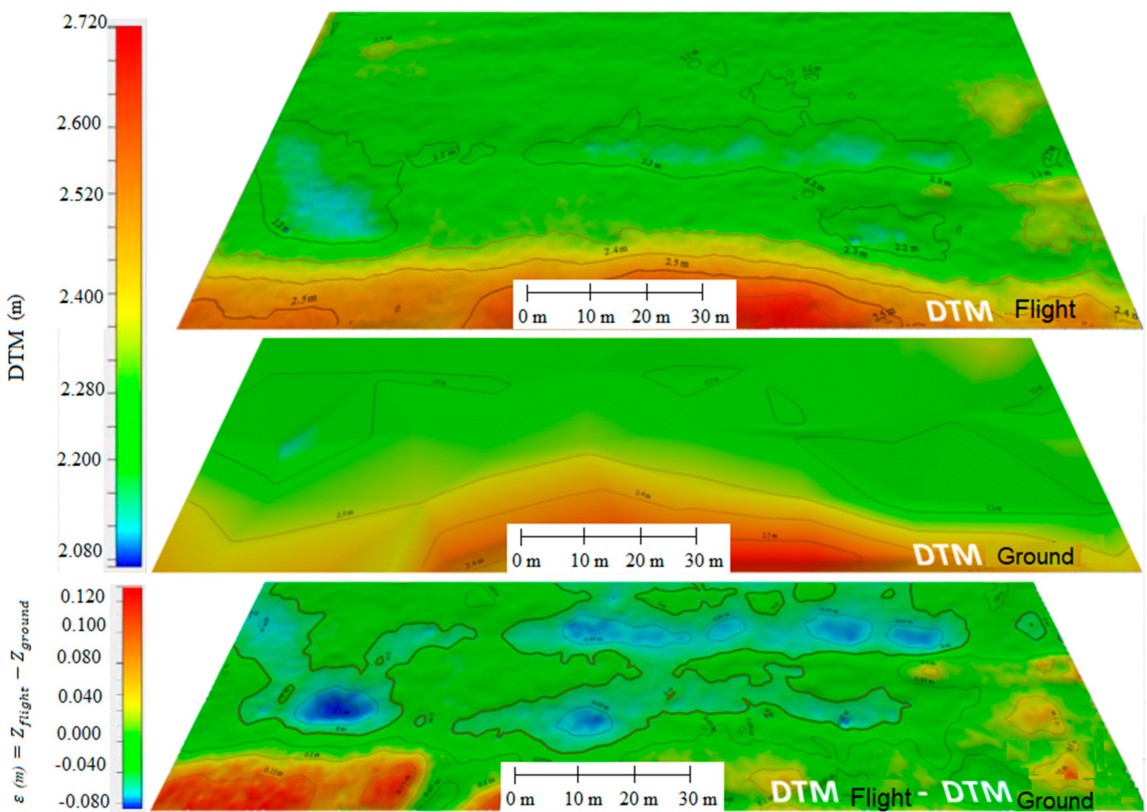

**Figure 6.** Map of the vertical error in every grid($\varepsilon$). Example of the difference between the altitudes of the flight digital terrain models (DTM) and the topographic DTM (GPS on the ground).

Given the higher precision in the horizontal plane (approximately twice that in the vertical plane) and the very gentle slope of the beach profile (<2%), we will assume that the influence of the possible location error of a point on the vertical precision is negligible.

However, this average of the vertical errors suffers from that positives and negatives can cancel each other out and give a false sense of accuracy. That is the reason why another statistic, the RMSE (Equation (2)) was calculated.

The National Standard for Spatial Data Accuracy (NSSDA) is a recent standard proposed by the Federal Geographic Data Committee (1998) [32] and can be used for both analog and digital cartographic data [33]. This standard assumes a normal distribution of $\varepsilon$ and uses the root-mean-square error (RMSE) as the most common and valid statistic for the evaluation of products obtained by photogrammetry and remote sensing.

$$RMSE_Z = \sqrt{\frac{1}{n}\sum_{i=1}^{n}\left(Z_{flight} - Z_{ground}\right)^2} \tag{2}$$

The 95% confidence interval (Equation (3)) for the vertical accuracy reached in each of the grids was determined according to the NSSDA as

$$P_{Z,95\%} = 1.96 \cdot RMSE_Z \tag{3}$$

Thus, Equation (4) shows the range of values that do not exceed the established accuracy.

$$\left\{ \begin{array}{c} \overline{x} + 1.96 \cdot RMSE_Z \\ \overline{x} - 1.96 \cdot RMSE_Z \end{array} \right\} \tag{4}$$

## 4. Results and Discussion

### 4.1. Error for Each of the 72 Cases

As previously mentioned, the average of the vertical errors ($\varepsilon$ in Equation (1)) results in a number not too helpful because positives and negatives can cancel each other out. That is the reason why another statistic, the RMSE (Equation (2)) was calculated. From these data, the vertical accuracy (Equation (3)) for each of the cases was determined. The results of these two values for each of the 72 cases are shown in Table 4. Moreover, another error parameter defined in the methodology is also presented in Table 4, the percentage of non-empty grids (1 × 1 m²), i.e., with at least one homogeneous point inside.

### 4.2. Influence of Number of GCPs

The first variable to consider is the number of GCPs. A box-and-whisker plot of their RMSE error is shown in Figure 7. Note that a boxplot is a standardized way of displaying the dataset based on a five-number summary: the minimum, the maximum, the sample median, and the first and third quartiles. You can see from the graph that there is a large distance between the lower (25%) and upper quartiles (75%) (IQR-Interquartile range), which are 0.10, 0.08, and 0.03 m for 5, 7, and 10 GCPs, respectively. Note that the whiskers (the two lines outside the box that extend to the highest and lowest observations) are similar in the three cases. The high value for the 10 GCP case is due to the existence of outliers for the noon survey. The variation of these results is far away from the results presented by other authors as James et al. [34] whose RMSE had a negligible deviation because of the number of GCPs, obtaining 3.12, 3.57, and 3.59 cm for 5, 10, and 15 GCPs, respectively.

**Table 4.** Different kinds of errors (RMSE, vertical accuracy, percentage of grids with homogeneous points) for data for different side (S) and forward (F) overlap and for different flight height, flight time, and number of GCPs.

| Flight Time | Flight Height (m) | Type of Error | Overlap | | | | | | | | | | | |
| --- | --- | --- | --- | --- | --- | --- | --- | --- | --- | --- | --- | --- | --- | --- |
| | | | 85%F−85%S | | | 85%F−70%S | | | 70%F−85%S | | | 70%F−70%S | | |
| | | | Number of GCPs | | | | | | | | | | | |
| | | | 10 | 7 GCP | 5 GCP | 10 GCP | 7 GCP | 5 GCP | 10 GCP | 7 GCP | 5 GCP | 10 GCP | 7 GCP | 5 GCP |
| 8 a.m. | 60 | RMSE (m) | 0.03 | 0.05 | 0.05 | 0.04 | 0.04 | 0.05 | 0.03 | 0.06 | 0.05 | 0.03 | 0.03 | 0.05 |
| | | Vertical accuracy (1.96·RMSE) | 0.06 | 0.10 | 0.10 | 0.08 | 0.08 | 0.10 | 0.06 | 0.12 | 0.10 | 0.06 | 0.06 | 0.10 |
| | | % grid with data | | 65.80% | | | 59.73% | | | 59.73% | | | 60.29% | |
| | 80 | RMSE (m) | 0.04 | 0.04 | 0.04 | 0.04 | 0.04 | 0.04 | 0.04 | 0.04 | 0.04 | 0.04 | 0.04 | 0.05 |
| | | Vertical accuracy (1.96·RMSE) | 0.08 | 0.08 | 0.08 | 0.08 | 0.08 | 0.08 | 0.08 | 0.08 | 0.08 | 0.08 | 0.08 | 0.10 |
| | | % grid with data | | 60.45% | | | 60.15% | | | 60.85% | | | 55.83% | |
| | 100 | RMSE (m) | 0.05 | 0.07 | 0.09 | 0.05 | 0.05 | 0.09 | 0.05 | 0.05 | 0.08 | 0.05 | 0.05 | 0.19 |
| | | Vertical accuracy (1.96·RMSE) | 0.10 | 0.14 | 0.18 | 0.10 | 0.10 | 0.18 | 0.10 | 0.10 | 0.16 | 0.10 | 0.10 | 0.37 |
| | | % grid with data | | 46.00% | | | 36.00% | | | 36.00% | | | 35.97% | |
| 12 p.m. | 60 | RMSE (m) | 0.06 | 0.05 | 0.05 | 0.10 | 0.14 | 0.15 | 0.033 | 0.04 | 0.04 | 0.14 | 0.15 | 0.16 |
| | | Vertical accuracy (1.96·RMSE) | 0.12 | 0.10 | 0.10 | 0.20 | 0.27 | 0.29 | 0.06 | 0.08 | 0.08 | 0.27 | 0.29 | 0.31 |
| | | % grid with data | | 43.24% | | | 39.26% | | | 37.60% | | | 38.53% | |
| | 80 | RMSE (m) | 0.05 | 0.05 | 0.05 | 0.10 | 0.14 | 0.16 | 0.05 | 0.05 | 0.05 | 0.145 | 0.17 | 0.17 |
| | | Vertical accuracy (1.96·RMSE) | 0.10 | 0.10 | 0.10 | 0.20 | 0.27 | 0.31 | 0.10 | 0.10 | 0.10 | 0.27 | 0.33 | 0.33 |
| | | % grid with data | | 39.03% | | | 36.17% | | | 36.95% | | | 33.47% | |
| | 100 | RMSE (m) | 0.06 | 0.12 | 0.13 | 0.07 | 0.04 | 0.09 | 0.06 | 0.07 | 0.07 | 0.15 | 0.18 | 0.19 |
| | | Vertical accuracy (1.96·RMSE) | 0.12 | 0.24 | 0.25 | 0.14 | 0.08 | 0.18 | 0.12 | 0.14 | 0.14 | 0.29 | 0.355 | 0.37 |
| | | % grid with data | | 29.66% | | | 20.47% | | | 21.37% | | | 19.18% | |

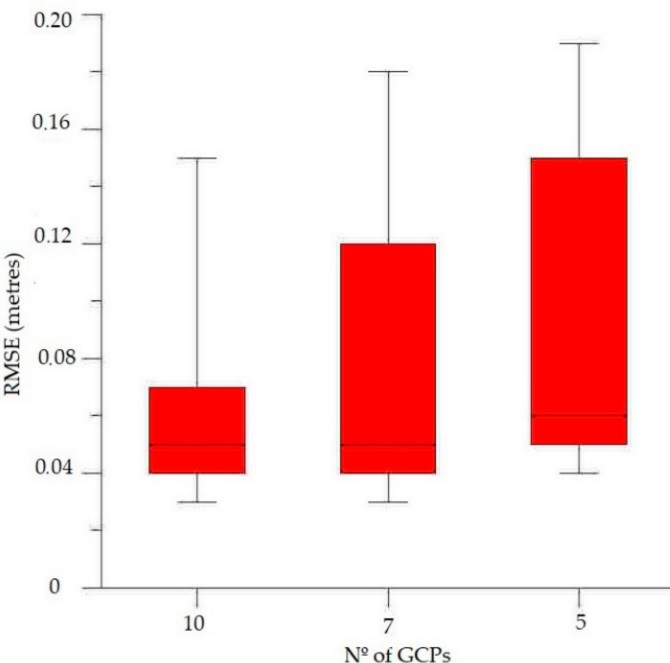

**Figure 7.** Box-and-whisker plot of the average error based on the GCP number.

Thus, though other authors such as Zimmerman et al. [30] found that (7 to 9) well-placed GCPs in the optimal configuration produced the same magnitude of error as using more (15) poorly placed GCPs, the only acceptable values in our case are those collected by using 10 GCPs, with an IQR less than 3 cm.

Moreover, 10 GCPs is just the maximum number of points for this particular case. To generalize this value and its use in any other case, the number of GCPs has been divided by the surface in hectares (Ha), a unit frequently used in topographic surveys. Thus, if we divide 10 GCPs by 1.5 Ha (15,000 m$^2$), we get a rounded value for the density of GCPs (7 GCP/Ha), a new starting parameter when designing a beach leveling campaign using RPAS. Regrettably, a limitation of this study is that we did not check whether the accuracy might even increase more by using more than 10 GCP, and, therefore, the trend of the inclusion of more GCPs remains unknown.

### 4.3. Influence of Flight Time

As previously noted, visual homogeneity of beach areas is one of the main problems for surveying using photogrammetric methods because of the reduction of homologous points among adjacent frames. Therefore, two different times for the flights were chosen (8 a.m. and 12 a.m.) to find out if shadows in the early morning (Figure 8) produce more homologous points than at noon and, thus, a decrease in the error committed. The number of tie points are presented in Table 5.

Based on the results of the previous subsection, the values obtained for 5 and 7 GCPs were discarded to avoid distorting the statistical results of the rest of the variables. Figure 7 shows the values of vertical RMSE for both aforementioned times of the flight and all the flight heights

As can be seen in Figure 8 the IQR ranges from 5 to 14 cm with a median of 0.09 m at noon while there are just a mean of 3.5 cm, no outliers, and a negligible dispersion in the early morning. Moreover, the percentage of grids with data is almost 60% higher for early flights than for noon flights. Therefore, it can be established that RPAS for beach leveling must be performed early in the morning. Or, in other words, RPAS surveys must be banned at noon.

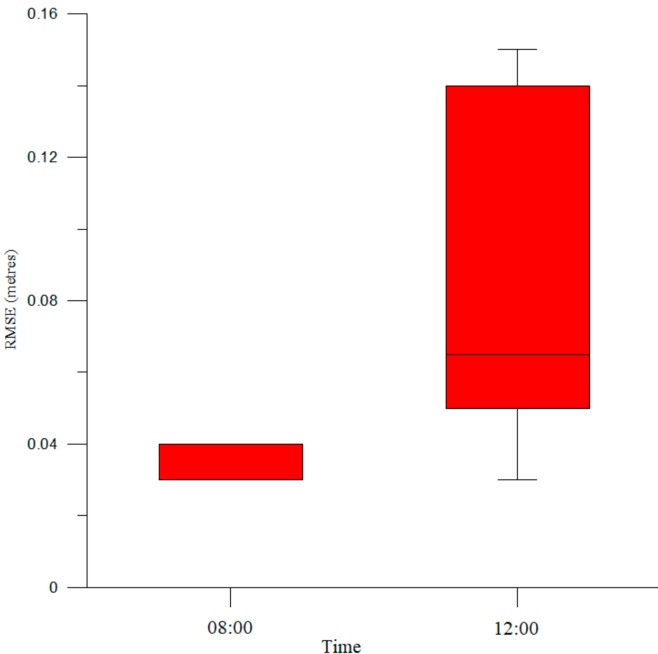

**Figure 8.** Box-and-whisker plot of RMSE vs. the flight time (8 a.m. and 12 a.m.) for all the flight heights.

**Table 5.** Number of tie points.

| Flight Time | Flight Height | Number of Tie Points |
|---|---|---|
| 08:00 a.m. | 60 m | 105,382 |
|  | 80 m | 76,852 |
|  | 100 m | 32,733 |
| 12:00 a.m. | 60 m | 39,296 |
|  | 80 m | 29,639 |
|  | 100 m | 19,848 |

### 4.4. Influence of Frame Overlap and Flight Height

Analyzing the percentage (70% vs. 85%) of side and forward overlap, four different cases were considered. Moreover, flights were carried out at three different heights (60, 80, and 100 m). The results presented in Table 4 are now shown in Figure 9, where only the 10 GCP experiments have been taken into account.

Starting with the noon flights (dashed lines), it can be seen that there is not too much difference between the results of 60 and 80 m flight heights. Regarding the forward overlap, there is no appreciable difference between both (70 and 85) percentages, i.e., forward overlap change from 85 to 70 did not influence final results. However, RMSE decreased enormously (from 0.15 to 0.05 m) when the side or transverse overlap changed from 70% to 85%.

On the other side, when results from 8 am flights are analyzed, RMSE remained constant and, therefore, independent from both side and forward overlap percentages. Furthermore, there is a small but still significant difference for RMSE as a function of the flight height. RMSE was 3, 4, and 5 cm for 60, 80, and 100 m heights, respectively. Following Gonçalves and Henriques [35], a relative accuracy of the flying height can be calculated. This relative accuracy was 0.35‰ in their case (from 0.046 m to 131 m flying height). A numerical value very similar to the results presented here where relative accuracy was 0.5‰ for each of the flights performed early in the morning but lesser than the values found for the flights performed at noon which can reach up to 1.5‰.

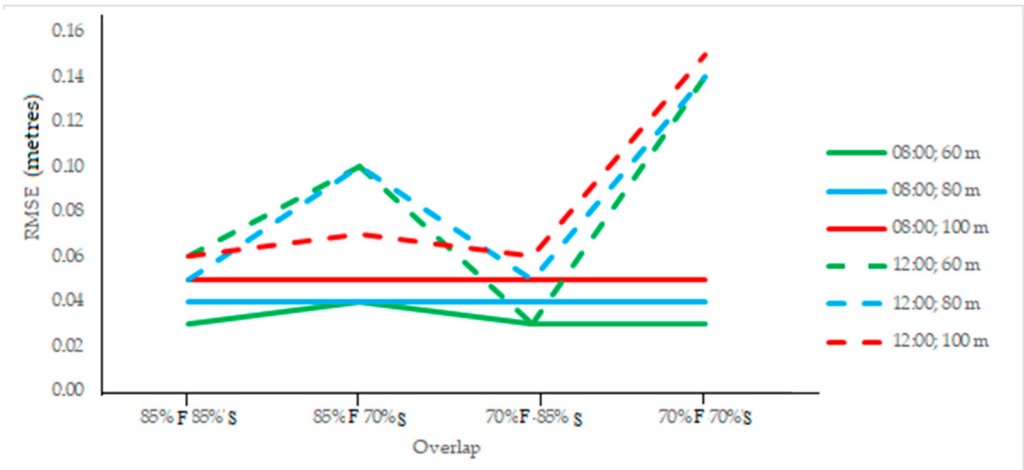

**Figure 9.** RMSE vs. different side (S) and forward (F) overlap for different flight heights and flight times (dashed lines are used for noon flights while solid lines are for 8 a.m. flights). Note that the number of GCPs is not a variable because only experiments performed with 10 GCPs were considered.

Thus, in brief, it can be established again that early morning flights minimize vertical error. Moreover, side overlap should not be less than 85% while forward overlap percentage is not a decisive factor. Finally, the decision about the flight height (when designing an RPAS for a beach leveling) must take into account that variation of vertical RMSE, though small in absolute value (5 cm vs. 3 cm), can be relatively significant (about 66%) when a 100 m height is chosen instead of a 60 m height.

## 5. Conclusions

A common fact in the photogrammetry of beaches (poorly differentiated visual areas) is the difficulty in the identification of common points in contiguous frames [22]. And, obviously, the fewer the homologous points defined by the program, the lower the accuracy. Thus, the main objective of this work is to determine the parameters of flight (height, time, frame overlap) and number of GCPs to optimize the accuracy of photogrammetric surveys when using RPAS in cases of visually homogeneous areas.

The following variables have been taken into account: flight height (60, 80, and 100 m), flight time (8 a.m. and 12 p.m.), side and forward overlap (70% vs. 85%), and the number of ground control points or GCPs. The combination of these variables results in 72 cases.

Firstly, one of the main conclusions is related to the density of GCPs. A minimum value of 7 GCPs/Ha has been found for this new parameter when designing a beach leveling campaign using RPAS. However, the trend of the inclusion of more GCPs remains unknown. This aspect is pending for future research.

Secondly, there is no appreciable difference regarding the forward overlap. But, on the other side, RMSE increased to three times (from 0.05 to 0.15 m) when side overlap decreased from 85% to 70%.

Moreover, the median of the error for noon flights (7 cm) is double that for the early morning flights (3.5 cm) because of the higher (almost 60%) percentage of grids with data for early flights. Therefore, beach levelings must never be performed at noon when carried out by RPAS.

Finally, there is a significant difference (till 66%) for RMSE as a function of the flight height. RMSE was 3, 4, and 5 cm for 60, 80, and 100 m heights, respectively, when only results from the 8 a.m. flights are analyzed. Furthermore, in this case, RMSE remains constant, and therefore independent, for the different side and forward overlap percentages.

**Author Contributions:** Conceptualisation, F.C.-d.-V.; F.J.G. and J.J.M.-P.; investigation and writing—original draft preparation, F.C.-d.-V.; F.J.G.; J.J.M.-P.; A.C.-d.-V.; V.R.-O.; P.L.; S.G.-L.; B.J.; review and editing, F.C.-d.-V.; J.J.M.-P. All authors have read and agreed to the published version of the manuscript.

**Funding:** This research was partially funded by Fundacion Campus Tecnologico de Algeciras. And the APC was funded by the Coastal Engineering Research group (University of Cadiz).

**Informed Consent Statement:** Not applicable.

**Data Availability Statement:** Data sharing not applicable.

**Conflicts of Interest:** The authors declare no conflict of interest.

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
