# Peer review of "Beach Leveling Using a Remotely Piloted Aircraft System (RPAS): Problems and Solutions"

_jmse, doi:10.3390/jmse9010019_

Round 1

Reviewer 1 Report

Please find detailed comments in attatched pdf file.

Reviewer 2 Report

This is a re-submitted version of a study which is focusing on the impact of different flight parameters on the quality of topographic derivates from UAV surveys. Even though the authors put a lot of effort in the revision and significantly changed the manuscripts the text still needs further improvement prior to publication in JMSE. Crucial information on the methods are still missing and parts of the results section are ambiguous and not very clear to the reader. Please find detailed points below:

  • In my perspective, the paper still contains too many definitions that should be taken for granted and which do not belong to a scientific journal paper as they are well-known in the community. E.g. GSD (line 189), EXIF (line198)
  • Decide on a term: side frame overlap (line 25) or side overlap (e.g. 159,…)
  • Structure: you introduce subsubsections but without a title (e.g.3.1.3 , 3.1.2, …). I would suggest you either add a title or you increase the granularity of depth (e.g. stay with 3.1 and 3.2)
  • Subsubsection 3.1.2: How many GPS points were collected to calculate the reference DTM? In my perspective, it is highly questionable if a DTM derived from a GPS survey is the representative reference. In a beach setting you usually see a lot of small-scale forms which can be represented by a DTM of the UAV but not by a DTM of a GPS. In your analysis it is not clear how you deal with this difference in the resolution and the level of detail of your derived DTM. Figure 6 nicely represents this difference and I expect that the majority of errors are caused by the interpolation of your GPS survey results as small-scale topographic features are not captured.
  • Figure 1: Here you schematically present different numbers of GCPs. However, it is not clear to me, how you actually distributed 10 GCPs as you include only 9 GCPs in your scheme. I would suggest to add the chosen GCP distribution to Figure 5.
  • Line 202 and 205: please replace the term link points by the correct photogrammetric term tie points
  • Line 213: What is the motivation to down-sample the DTM to a 1m grid? You lose a lot of information and the validity of your results is minimized enormously. Working with an average value does not always reveal changes even though they might be apparent (e.g. positive and negative derivations from the original grid still bring you the same average and can only be visualized by the standard deviation)
  • Line 226: What does this reliability parameter affect the results? Which quality criterion is affected by the number of grids including a “datum”? Do you refer to a GPS measurement here?
  • Figure 6: Currently it is impossible to read the scale of the DTM. Please enlarge the scalebar.
  • Table 2: Please reduce the information to the configurations in your flight (e.g. you didn’t utilize a video, so you don’t need to show information on the ISO range for videos, only include the image size you actually considered)
  • Line 352: I don’t understand this sentence. It is not clear in which aspect no appreciable difference was observed.
  • I cannot find limitations of the study and an outlook for future research. I strongly encourage the authors to add these crucial parts to their manuscript.

Round 2

Reviewer 2 Report

Thank you for revising the manuscript and taking my comments into consideration. I have one more comment before the manuscript can go to production. In you answer, you mention the method for taking the GNSS RTK measurements to create the reference DTM (i.e. a total number of 657 survey points...). I think this information on the absolute number of survey points and their density (average point distance) is crucial for the reader and should be mentioned in the manuscript.

Author Response

Answer to Reviewer 2 (Round 2)

Thank you for revising the manuscript and taking my comments into consideration. I have one more comment before the manuscript can go to production. In you answer, you mention the method for taking the GNSS RTK measurements to create the reference DTM (i.e. a total number of 657 survey points...). I think this information on the absolute number of survey points and their density (average point distance) is crucial for the reader and should be mentioned in the manuscript.

Thank you very much to the reviewer for his/her comment.

A new sentence has been added. New lines 146-147

Thank you very much for your kind and thorough revision

This manuscript is a resubmission of an earlier submission. The following is a list of the peer review reports and author responses from that submission.

Round 1

Reviewer 1 Report

Please find attatched detailed review.

Reviewer 2 Report

The manuscript presents an interesting study on the use of RPAS for beach modelling. Different flight parameters were chosen and resulting DTMs evaluated and compared against GNSS observations. The recommendation that flights should not be carried out during noon is highly valuable and of particular interest to scholars monitoring coastal zones. Although the manuscript is well presented, serious flaws are prevalent in all sections.  Firstly, the authors do not discuss any research related to flight parameters and the quality assessment of DTMs generated from images of RPAS. During the past years, more than 50 papers were published discussing various GCP configurations and flight configurations – but the manuscript does not reference and discuss this work.  Secondly, vital information of the study setup is missing, such as the brand and type of the RPAS, number and distribution of measured GCPs, image processing settings as well as information on how the GNSS-based DTM was generated. Without outlining those essential facts, the interpretation of the results is not possible, e.g. it is not clear how many checkpoint residuals were used to evaluate the RMSE and vertical error in Table 3.  Thirdly, two of the so-called solutions cannot be supported by the findings presented in the paper. The authors claim to recommend 7 GCPs/ha as their results of the case with 10 GCPs suggests the best results. However, the authors did not check whether the accuracy might even increase more by using more than 10 GCP. I see this very critical to claim such a finding without knowing the trend of the inclusion of more GCPs. Additionally, the authors highlight the error function with an increase of the flight height. Although this might be partly true, the central fact of this comparison is not the flight height, but the GSD, which is proportional to the flight height. Thus, the vertical error should be presented in relation to the GSD. Lastly, the results are not adequately discussed, and a critical reflection on the finding is missing entirely.

Overall, the impressions outlined above lead to my suggestion to reject this manuscript. Since it is still a highly interesting subject and a good dataset of RPAS-based images, I encourage re-submission after a complete revision.

Reviewer 3 Report

Dear authors,

Please find my comments in the attached pdf. Overall, you need to strengthen the interpretation part. Please thoroughly review your manuscript with respect to the literature/software recommendations in the pdf's comments.
